# Determinants of Future Physical Activity Participation in New Zealand Adolescents across Sociodemographic Groups: A Descriptive Study

**DOI:** 10.3390/ijerph20116001

**Published:** 2023-05-30

**Authors:** Tom Bergen, Alice Hyun Min Kim, Anja Mizdrak, Louise Signal, Geoff Kira, Justin Richards

**Affiliations:** 1Department of Public Health, University of Otago, Wellington 6242, New Zealand; anja.mizdrak@otago.ac.nz (A.M.); louise.signal@otago.ac.nz (L.S.); 2Sport New Zealand Ihi Aotearoa, Wellington 6011, New Zealand; justin.richards@vuw.ac.nz; 3Biostatistics Group, Dean’s Department, University of Otago, Wellington 6242, New Zealand; alice.kim@otago.ac.nz; 4Te Hau Kori, Faculty of Health, Victoria University of Wellington Te Herenga Waka, Wellington 6012, New Zealand; geoff.kira@vuw.ac.nz

**Keywords:** physical activity, adolescent, young people, sport, exercise, leisure, behaviour, demographic, intervention, well-being

## Abstract

This cross-sectional study aimed to explore various determinants of future physical activity (PA) participation in adolescents across sociodemographic groups. Sociodemographic characteristics (age, gender, ethnicity, deprivation status, physical disability status) were assessed in a national sample (*n* = 6906) of adolescents (12–17 years old) between 2017 and 2020 in New Zealand. The determinants of future PA participation chosen for analysis included current indicators of PA participation (i.e., total time, number of types, number of settings). We also examined widely recognised modifiable intrapersonal (i.e., physical literacy) and interpersonal (i.e., social support) determinants of current and future PA behaviour, along with indicators of PA availability issues. Older adolescents scored worse across all determinants of future PA than younger adolescents, with a key transition point appearing at 14–15 years of age. Māori and Pacific ethnicities scored best across each determinant category on average, with Asian populations scoring the worst. Gender diverse adolescents scored substantially worse than male and female adolescents across every determinant. Physically disabled adolescents scored worse than non-disabled across all determinants. Adolescents from medium and high deprivation neighbourhoods scored similarly across most determinants of future PA participation and both tended to score worse than people from low deprivation neighbourhoods. A particular focus on the improvement of future PA determinants is warranted within adolescents who are older, Asian, gender diverse, physically disabled, and from medium to high deprivation neighbourhoods. Future investigation should prioritise the longitudinal tracking of PA behaviours over time and develop interventions that affect multiple future PA determinants across a range of sociodemographic backgrounds.

## 1. Introduction

Insufficient physical activity (PA) participation across the lifespan is a worldwide public health priority, including in Aotearoa, New Zealand (NZ) [1,2]. Adolescence, specifically, is an important life stage in which the social determinants of health must be monitored and influenced appropriately to ensure lifelong benefits [3]. Regular PA participation is well documented to provide numerous health and wellbeing benefits [4,5,6,7] and interventions focused on improving PA in adolescents can be highly effective at improving public health on a large scale [8,9]. A growing body of international evidence supports intervention during adolescence to create positive lifelong changes in PA behaviours by addressing key determinants of current and future participation [10,11]. Therefore, identifying the PA determinants relevant to adolescents and their differences across different sociodemographic groups may prove useful for promoting future participation and lifelong wellbeing benefits.

It is widely accepted that no single factor explains how PA participation levels change over time, and there is extensive literature relating to the determinants that predict whether adolescents will be active or not [12,13]. However, few studies look at what factors during adolescence influence future PA participation. Additionally, very few longitudinal studies follow the same adolescents for a significant amount of time across their lifespans to see the full extent of how their adolescent behaviour and environment influences their future PA participation [14]. PA behaviour is also known to vary according to several socio-demographic factors (e.g., age, gender, ethnicity). The combination of the above complexities presents a challenge when deciding on the best way to intervene to promote future PA within different adolescent population groups.

One approach to finding relevant determinants of future PA participation is to examine the current state of PA behaviours in adolescents. Longer durations of weekly PA participation in adolescence appear to translate to higher levels of PA in adulthood [15,16,17]. However, duration is not the sole determinant of adult PA; the number of settings or types of PA adolescents participate in also matters [18,19]. Furthermore, the socio-ecological model allows us to explore the complex relationships between the different societal factors surrounding an individual or group and is a useful tool for promoting positive behaviour change [20]. There is an abundance of evidence detailing the impact of various factors across the socio-ecological model on adolescent and adult PA behaviours. For example, intrapersonal PA determinants include physical literacy, which refers to the knowledge, motivation, confidence, and competence to be active throughout the lifespan [21,22]. Interpersonal determinants, including social support, are also important, as the teachers, peers, coaches, and family around an adolescent can occupy many roles related to motivating and teaching PA [23,24]. The physical environment that surrounds an adolescent, such as the actual and perceived availability of PA opportunities, also influences PA behaviours [25].

Examining determinants of PA may help identify potential levers for promoting future participation. This study aimed to explore and describe the current state of PA participation in NZ adolescents (aged 12–17) expressed through various determinants of future PA across sociodemographic groups. We intend our results to inform policymakers and practitioners in the development of more targeted and relevant interventions for promoting PA in adolescents.

## 2. Materials and Methods

### 2.1. Study Design, Participants, and Procedures

The Active NZ survey is a nationally representative cross-sectional survey conducted by Sport New Zealand Ihi Aotearoa. The Active NZ Young People’s survey is run adjunct to the Active NZ adult survey. Recruitment of adults to the Active NZ survey is conducted via a household-level sampling strategy. This strategy uses the NZ electoral roll as a sampling frame to identify adult respondents. In turn, young people aged between 5- and 17-years old residing in those same households of adult respondents of the adult survey are recruited. A complex survey design with household clusters and stratification by region, ethnicity, and age group is used to achieve a nationally representative random sample across key sociodemographic backgrounds. The Active NZ Young People’s survey records a wide range of information regarding PA behaviours and attitudes in addition to sociodemographic information. We accessed data from the 2017–2020 waves for our analyses. Further information about the Active NZ survey methods can be obtained from the annual technical reports [26,27,28].

### 2.2. Sociodemographic Variables

Key sociodemographic variables captured in the survey are described below.

Age: Respondent age was recorded in years and analysed as a categorical variable with levels 12 to 17.

Gender: Respondents self-identified their gender as male, female, or gender diverse.

Ethnicity: Respondents were allowed to select multiple ethnic groups that they identified with from a large selection of ethnic categories. This was to reflect the multicultural context and to ensure that the survey analysis is culturally relevant and responsive. The responses were converted to a series of binary variables to record each ethnic group’s identification. The analyses were conducted on the following variables: Māori, European, Pacific, Asian, and Other. The “Other” category was used to combine less represented ethnic groups. A person can be included as being of multiple ethnic backgrounds at the same time, e.g., Māori and European; hence, the percentages identifying with these groups sum over 100%.

Physical disability status: Respondents who reported using a wheelchair, using a walking aid, using prosthetics, or dealing with an ongoing physical illness were classified as having a disability.

Neighbourhood deprivation status: Deprivation was determined using the 2018 NZ Index of Deprivation, which combines census data relating to income, home ownership, employment, qualifications, family structure, housing, access to transport, and communications to designate small geographic areas (called the census meshblocks, generally containing between 100 and 200 people) with a decile number ranging from 1 (least deprived) to 10 (most deprived) [29]. The survey records were matched to census meshblocks using the domicile information. Respondents were classified as residing in low (deciles 1–3), medium (deciles 4–7), and high (deciles 8–10) deprivation areas.

### 2.3. Key Determinants of Future PA Participation

The determinants of future PA participation analysed in this study are briefly summarised below.

Current PA participation: Respondents provided data for three variables (current PA duration, settings, and types) that characterised their current PA:Current weekly PA duration—Respondents were asked to identify what activities they participated in during the past seven days and how long they participated in each for. The survey listed 77 activities and provided an “Other” option [26,27,28]. The total sum of these durations was calculated as a numerical value for this variable.Current number of PA settings—For activities that they had participated in, respondents were asked to answer yes/no to what settings they had participated: “In PE or class at school” (physical education); “In a competition or tournament” (competitive sport); “Training or practising with a coach/instructor” (coached sports training); “Playing or hanging out with family or friends” (social sport); “Playing on my own” (solo sport); “For extra exercise, training, or practice without a coach or instructor” (uncoached sports training). The total number of settings that they had participated in determined a final score for this variable ranging from 0 to 6.Current number of PA types—From the 77 PA options provided, we developed a summated score showing the number of activities that any one respondent participated in.Current physical literacy: An aggregated measure of responses to four questions regarding knowledge, confidence, competence, and motivation to participate in PA. Full details of the questions are provided in Appendix A. The response for each question ranged from 1 to 5 (1 = Disagree a lot, 5 = Agree a lot). The final score ranged from 4 (very low) to 20 (very high).Current social support for PA: An aggregated measure of responses to eleven questions that were grouped into five score categories: (1) “Family/peer social barriers” (Questions 1–4); (2) “General social barriers” (Questions 5–8); (3) “Social encouragement” (Question 9); (4) “Social cohesion” (Question 10); and (5) “Family enjoyment” (Question 11). Full details of the questions are provided in Appendix A. The response for each question ranged from 1 to 5 (Questions 1–8: 1 = Disagree a lot, 5 = Agree a lot; Questions 9–11: 1 = Little social support, 5 = Great social support). The final score ranged from 5 (very low) to 25 (very high).Current PA availability issues: An indicator of whether PA is perceived as available to adolescents and whether they have the capability to be active in the way they want. This binary variable was constructed from the responses to two questions. The first question asked “What activities have you participated in during the last 7 days?”. Participants then had the option of stating whether they wanted to perform more PA or not. Of those that wanted to perform more PA, a second question was asked “What’s ONE activity you would do more of if you could?”. Some participants wanted to perform more of an activity they did not currently participate in (attained from question 1), so we assumed they had availability challenges (coded as yes). Those that wanted to perform more of a PA they were already participating in were assumed to not have availability challenges (coded as No). The final score was presented as the proportion of respondents that had availability issues.

### 2.4. Statistical Analyses

The summary statistics for each variable were computed for the raw responses and in a weighted analysis obtaining the national estimates for these variables. Numeric scores, such as total PA or physical literacy scores, were summarised into means and standard deviations for unweighted data and survey means and standard errors in a weighted analysis. Categorical variables, such as the indicator for having availability issues, were summarised using the number of respondents and percentages for the unweighted dataset in Table 1 and as an estimate of population proportion (expressed as percentages %), at 95% confidence intervals (CI), for weighted analyses in Table 2.

The weighted national estimates for survey means and percentages for each variable were computed using the survey package v4.0 [30] in R (R Statistical Foundation, Vienna, AT) using the RStudio interface (2022.02.1, build 461). The weights in the survey were adjusted using the Iterative Proportional Fitting (IPF) technique, which incorporates known population data on sociodemographic information such as the ratio of people to total population in each district by gender and ethnic group. A more detailed description of survey design and the implementation of IPF is published elsewhere [26,27,28]. The CIs were computed using the cluster robust estimators based on the linearisation method.

To graphically represent the average differences in each variable across sociodemographic groups and to display the variables in a single plot, each variable was standardised using its overall mean and standard deviation. The weighted averages and 95% CIs were computed for the standardised scores and then plotted in Figure 1, Figure 2, Figure 3, Figure 4 and Figure 5.

## 3. Results

### 3.1. Participant Characteristics

Participant characteristics are reported in Table 1. Data on *n* = 6906 NZ youth aged 12–17 years were used in the analysis. A higher proportion of younger age groups participated in the survey, with more than 40% being 12–13 years of age. Female respondents accounted for the majority while a small percentage (<1%) identified as gender diverse. Most young people that were surveyed identified as having a European ethnic background, followed by Māori, Asian, Pacific, then all other ethnicities. Only a small proportion of respondents reported having one or more physical disabilities. A similar number of survey participants were from low or medium deprivation neighbourhoods, with fewer respondents from high deprivation neighbourhoods.

**Table 1 ijerph-20-06001-t001:** Number and proportion of participants by sociodemographic characteristics.

Sociodemographic Variables		n (%)
**Age (years)**		
12	1477	(21.4)
13	1333	(19.3)
14	1258	(18.2)
15	1137	(16.5)
16	944	(13.7)
17	757	(11.0)
**Gender**		
Male	3003	(43.5)
Female	3853	(55.8)
Diverse	50	(0.7)
**Ethnicity**		
Māori	1008	(14.6)
European	5845	(84.6)
Pacific	332	(4.8)
Asian	754	(10.9)
Other	165	(2.4)
**Physical Disability**		
Yes	389	(5.6)
**Deprivation Status ^a^**		
Low (1–3)	2539	(36.9)
Mid (4–7)	2247	(32.7)
High (8–10)	1004	(14.6)
Unknown	1091	(15.9)

^a^ *n* = 25 (0.4%) missing.

### 3.2. Current Determinants of Future PA across Sociodemographic Groups

The overall and stratified results for the assessed indicators of future PA determinants are reported in Table 2.

#### 3.2.1. Age

Point estimates of all measures of PA participation, physical literacy, and social support are worse in older adolescents compared to younger adolescents. For example, the estimate of the population average for current weekly PA duration for 12-year-old adolescents was 13.4 (95% CI: 12.6, 14.3), which was higher than that of 17-year-old adolescents, whose average was 7.4 (95% CI: 6.6, 8.2). Although a similar pattern was observed for activity availability, 12-year-old adolescents had marginally worse scores than 13–14-year-olds, and it is only at the age of 15 years that substantial differences are apparent.

#### 3.2.2. Gender

Males scored better than females across all reported determinants of future PA participation. The point estimates for gender diverse respondents were lower than both male and female respondents across all determinant scores but had wide CI estimates (widths ranging 0.9 to 5.7).

#### 3.2.3. Ethnicity

Māori scored highly across all determinants but particularly for total current weekly PA duration and current number of PA types. Pacific populations also scored highly across most determinants, with the point estimate for their current social support score being the highest of all ethnicities, along with having the lowest score for current PA activity availability. Asian adolescents appeared to score worse than all other ethnicities across every determinant of future PA participation. Ethnicities in the “Other” group displayed varying results, but, notably, appeared to score the best for current physical literacy and current number of PA settings.

#### 3.2.4. Physical Disability Status

Those living with a disability had worse point estimates than those without disabilities across every determinant of future PA participation. However, the confidence intervals were wider for disabled respondents than for non-disabled respondents.

#### 3.2.5. Deprivation

The point estimates of adolescents with low deprivation levels were better than those with medium and high deprivation across all determinants. Other than PA duration, there is little difference between determinant scores for adolescents with medium and high deprivation levels.

**Table 2 ijerph-20-06001-t002:** Average determinant scores across adolescents categorised by sociodemographic characteristics (*n* = 6906).

Sociodemographic Variables	Current PA Participation			Current Physical Literacy	Current Social Support for PA	Current PA Availability Issues
	Current Weekly PA Duration (Hrs/Week)	Current Number of PA Settings	Current Number of PA Types			
	**Average Score (95% CI)**					**Percentage (95% CI)**
** *Overall* **	10.8 (10.5, 11.1)	3.3 (3.2, 3.3)	5.5 (5.40, 5.6)	16.6 (16.5, 16.7)	21.4 (21.3, 21.4)	58.3 (56.5, 60.0)
** *Missing n (%)* **	34 (0.5%)	258 (3.7%)	0 (0%)	957 (13.9%)	957 (13.9%)	2144 (31.0%)
** *Age (yrs)* **						
12	13.4 (12.6, 14.3)	3.6 (3.5, 3.7)	7.3 (7.0, 7.6)	17.1 (16.9, 17.3)	21.9 (21.7, 22.1)	56.2 (52.2, 60.2)
13	12.1 (11.4, 12.9)	3.5 (3.4, 3.6)	6.4 (6.2, 6.7)	16.9 (16.7, 17.1)	21.8 (21.6, 21.9)	54.9 (51.0, 58.8)
14	10.8 (10.2, 11.5)	3.4 (3.3, 3.5)	5.6 (5.4, 5.9)	16.9 (16.7, 17.1)	21.4 (21.2, 21.6)	55.0 (50.7, 59.2)
15	9.9 (9.1, 10.6)	3.1 (3.0, 3.2)	4.7 (4.4, 4.9)	16.3 (16.0, 16.5)	21.1 (20.9, 21.2)	58.7 (54.5, 62.9)
16	8.5 (7.7, 9.4)	2.9 (2.8, 3.0)	4.0 (3.7, 4.2)	16.1 (15.9, 16.4)	20.7 (20.5, 20.9)	60.4 (55.8, 64.9)
17	7.4 (6.6, 8.2)	2.6 (2.5, 2.8)	3.5 (3.2, 3.8)	16.1 (15.8, 16.4)	20.8 (20.6, 21.1)	69.3 (64.3, 74.3)
** *Gender* **						
Male	11.7 (11.2, 12.2)	3.3 (3.3, 3.4)	5.7 (5.5, 5.9)	16.9 (16.7, 17.0)	21.6 (21.5, 21.7)	51.7 (49.1, 54.4)
Female	9.9 (9.5, 10.3)	3.2 (3.2, 3.3)	5.4 (5.2, 5.5)	16.4 (16.3, 16.6)	21.2 (21.1, 21.3)	63.6 (61.4, 65.9)
Diverse	8.8 (6.0, 11.7)	2.9 (2.5, 3.4)	5.1 (3.9, 6.4)	14.5 (13.2, 15.8)	19.2 (18.3, 20.1)	75.6 (59.0, 92.2)
** *Ethnicity* **						
Māori	12.4 (11.4, 13.3)	3.4 (3.3, 3.5)	6.0 (5.7, 6.3)	16.9 (16.6, 17.2)	21.6 (21.4, 21.8)	55.0 (50.4, 59.6)
European	11.0 (10.7, 11.3)	3.3 (3.3, 3.4)	5.7 (5.5, 5.8)	16.7 (16.7, 16.8)	21.4 (21.3, 21.5)	57.8 (56.1, 59.5)
Pacific	11.8 (10.3, 13.3)	3.4 (3.2, 3.6)	5.8 (5.2, 6.3)	17.0 (16.6, 17.4)	21.7 (21.4, 22.0)	54.4 (46.7, 62.1)
Asian	8.0 (7.3, 8.7)	3.0 (2.8, 3.1)	4.5 (4.1, 4.8)	15.8 (15.5, 16.1)	20.6 (20.4, 20.9)	61.3 (56.3, 66.3)
Other	11.3 (9.5, 13.0)	3.6 (3.3, 3.9)	5.2 (4.5, 5.8)	17.1 (16.7, 17.6)	21.3 (20.8, 21.7)	56.1 (46.2, 66.0)
** *Disability* **						
Non-disabled	10.8 (10.5, 11.1)	3.3 (3.2, 3.3)	5.5 (5.4, 5.7)	16.7 (16.6, 16.8)	21.4 (21.3, 21.5)	57.9 (56.1, 59.7)
Disabled	10.4 (9.2, 11.7)	3.1 (3.0, 3.3)	5.2 (4.7, 5.7)	15.9 (15.5, 16.3)	20.6 (20.3, 20.9)	64.6 (57.8, 71.4)
** *Deprivation Status* **						
Low (1–3)	11.0 (10.5, 11.5)	3.3 (3.3, 3.4)	5.6 (5.4, 5.8)	16.9 (16.7, 17.0)	21.4 (21.3, 21.6)	55.7 (52.9, 58.4)
Mid (4–7)	10.6 (10.1, 11.2)	3.3 (3.2, 3.3)	5.4 (5.2, 5.6)	16.4 (16.2, 16.6)	21.2 (21.1, 21.4)	59.9 (56.8, 62.9)
High (8–10)	10.4 (9.5, 11.3)	3.3 (3.1, 3.4)	5.4 (5.1, 5.7)	16.4 (16.1, 16.7)	21.3 (21.1, 21.5)	59.9 (54.8, 64.9)

### 3.3. Standardised Differences between Current Determinants of Future PA across Sociodemographic Groups

The standardised differences for each of the assessed indicators of future PA determinants are presented in Figure 1, Figure 2, Figure 3, Figure 4 and Figure 5.

#### 3.3.1. Age (Standardised)—Figure 1

Except for 14-year-olds, the determinant with the largest relative difference to the mean was current number of PA types for all other ages. Current physical literacy had the smallest relative difference to the mean at either end of the age spectrum (i.e., 12- to 13-year-olds and 16- to 17-year-olds). At age 14, the CIs for current PA duration, PA types, and social support contained the overall cohort mean. This may indicate an inflection point at age 14–15 years where relative determinant scores switch from being above the mean in younger age groups to below the mean in older age groups.

**Figure 1 ijerph-20-06001-f001:**
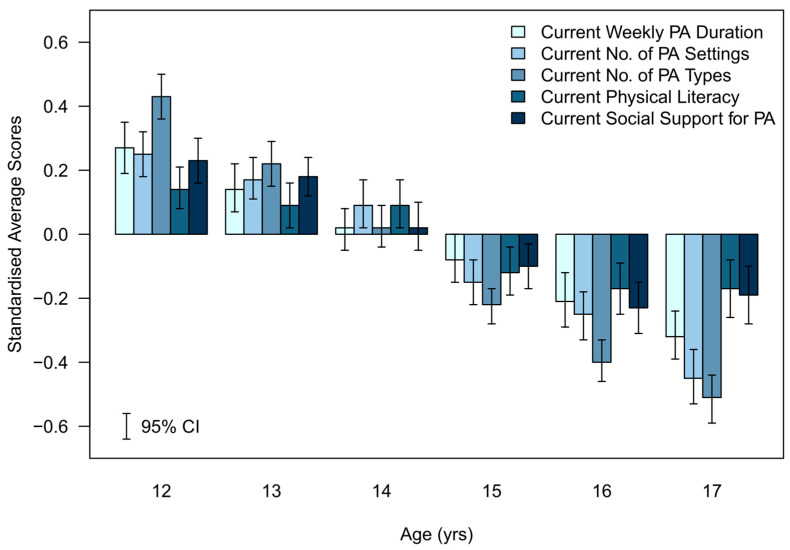
Standardised average scores of the determinants of future PA participation by age group.

#### 3.3.2. Gender (Standardised)—Figure 2

The relative difference from the mean was small across all determinants for both males in a positive direction and females in a negative direction. The magnitude of the relative difference from the mean was much larger and negative in direction across all determinants for gender diverse adolescents, particularly for current physical literacy and current social support for PA. However, the CIs were much wider for gender diverse adolescents.

**Figure 2 ijerph-20-06001-f002:**
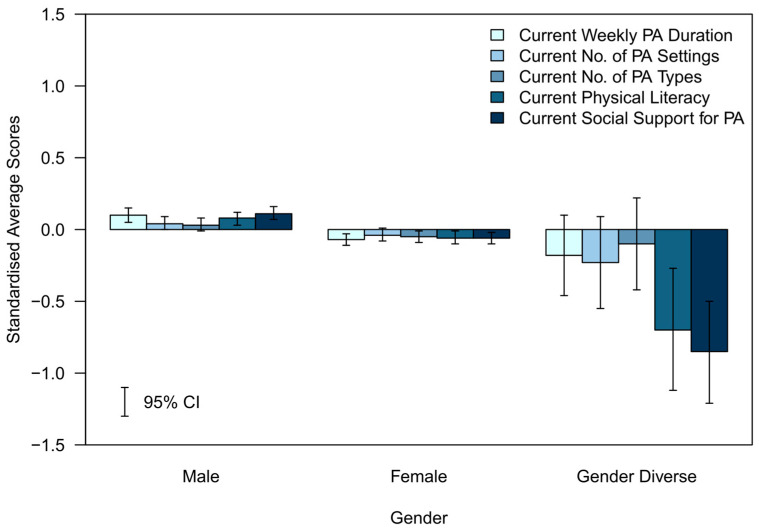
Standardised average scores of the determinants of future PA participation by gender.

#### 3.3.3. Ethnicity (Standardised)—Figure 3

The relative difference from the mean for all determinants was in a negative direction for the Asian group and in a positive direction for the European, Māori, and Pacific groups. Whilst there was consistency in the magnitude of the difference from the mean across all determinants for the European and Asian groups, this was not the case for the rest of the population. For the Māori respondents, the largest relative difference to the mean was for current weekly PA duration, but the results across the other determinants were reasonably consistent. The Pacific group reported large relative differences from the mean for current social support for PA and physical literacy, but there was a much smaller difference from the mean for current number of PA types. The pattern for the “Other” ethnicities group showed more variation with the relative difference for the current number of PA settings and current physical literacy, with both being substantially higher than the mean, while the current number of PA types appeared below the mean.

**Figure 3 ijerph-20-06001-f003:**
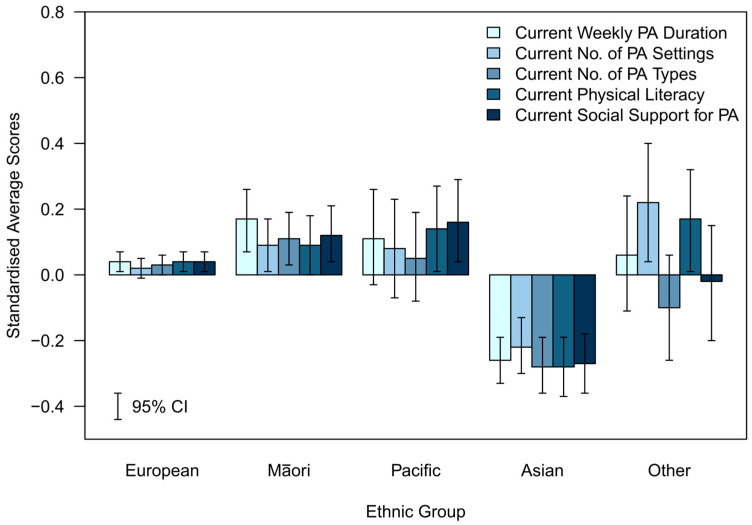
Standardised average scores of the determinants of future PA participation by total ethnic group.

#### 3.3.4. Physical Disability Status (Standardised)—Figure 4

The relative difference from the mean was small and in a positive direction across all determinants for non-disabled adolescents. The magnitude of the relative difference from the mean was much larger and negative in direction across all determinants for disabled adolescents, particularly for current physical literacy and current social support for PA. However, the CIs were much wider in disabled adolescents.

**Figure 4 ijerph-20-06001-f004:**
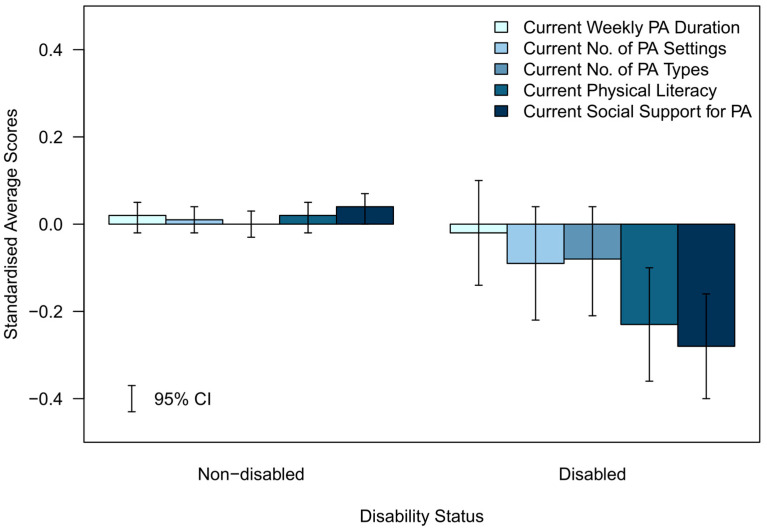
Standardised average scores of the determinants of future PA participation by disability status.

#### 3.3.5. Deprivation (Standardised)—Figure 5

In the low deprivation group, all determinants were in a positive direction from the mean, particularly for current physical literacy and current social support for PA, which both had CIs above the overall cohort mean. Contrastingly, current number of PA types had a small relative difference to the mean in the low deprivation group. In the medium and high deprivation groups, the relative difference from the mean was in a negative direction for all determinants of future PA. Both medium and high deprivation levels displayed lower current physical literacy scores than the mean, with the CI for the medium deprivation group being fully below the mean. The relative differences in the point estimates to the overall cohort mean were smaller for all other determinants of future physical activity in both the medium and high deprivation groups.

**Figure 5 ijerph-20-06001-f005:**
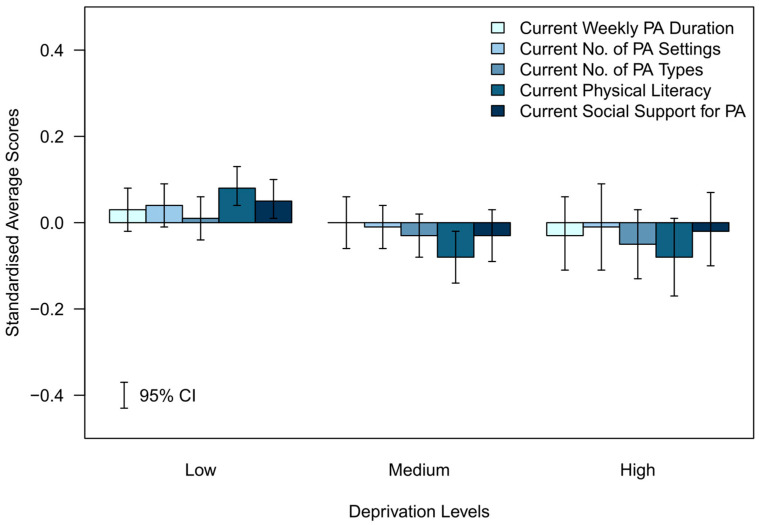
Standardised average scores of the determinants of future PA participation by deprivation levels.

## 4. Discussion

Our results indicate that there are disparities in the determinants of future PA among adolescents from different sociodemographic groups. Indicators for the determinants of future PA are worse in later adolescence, particularly for variety of participation types, and there appears to be a critical transition point that occurs at 14–15 years of age. Gender diverse adolescents scored worse than male and female adolescents across every determinant, particularly for indicators of current physical literacy and social support. Asian adolescents performed the worst of any ethnicity across every determinant, and there was minimal variation in the relative scores across all determinants for this group. Māori and Pacific populations performed well across all determinants, with Māori scoring highest in current total weekly PA and Pacific scoring highly for current physical literacy and social support. Disabled adolescents scored worse than able-bodied adolescents in all the reported determinants, particularly for current physical literacy and social support. Adolescents with medium or high deprivation have very similar scores across all reported determinants of future PA participation, and both display much worse physical literacy scores than those living with low deprivation.

Our results are consistent with previous research about the differences in PA behaviour from early to late adolescence [31,32]. On an interpersonal level, adolescents typically become more independent from their immediate family over time, which often coincides with an increase in peer influences [33]. Social norms affecting these peer groups place increasing priority on academic achievement at school and undertaking paid work as adolescents get older [31]. This displaces time previously used to be physically active, which is further compromised by optional participation in physical education within schools after the age of 15 years in NZ [34]. It is also during this period that the focus of organised sport often transitions from “participating” to “winning” and the cost of sports club fees and equipment increases [35]. Finally, active transport behaviours change as older adolescents are expected to secure their own travel to school and start to become independent car drivers. Together, these actual and perceived barriers are thought to explain the reprioritisation of PA and the decrease in participation opportunities commonly observed during adolescence and observed in our study [31].

Our results align with previous work about males performing better than females across a range of PA-related categories [36]. However, the difference between male and female determinants was small, which may reflect the success of national endeavours to increase female PA participation [37]. Research on gender diverse youth, while limited, shows that inclusive environments can facilitate positive PA habits, whereas discriminatory environments can provide barriers to further PA [38]. Our results indicate gender diverse youth are often in discriminatory PA environments that lower their confidence and motivation to be active, contributing to their low physical literacy scores. Furthermore, the low social support scores in gender diverse adolescents we observed may stem from a lack of social acceptance within PA environments [39].

Our finding that Māori adolescents engage in the most total PA weekly is consistent with past work exploring how PA interventions were effective approaches to improving the existing wellbeing inequities compared to European populations [40,41]. Pacific peoples traditionally conduct PA incorporating family involvement and utilising the land around them, which may explain the high levels of social support and PA availability observed for this population group in our study [42]. Contrastingly, the low determinant scores of Asian adolescents indicate major barriers to promoting PA nationally. For example, Asian populations may find it difficult adjusting to a new land and lifestyle (for new migrants), resulting in having few traditional PA options available and limiting their motivation to be active [43]. Additionally, the social norms of Asian populations often centre around prioritising other factors over PA, such as academic success [44]. Adolescents classified as "Other" ethnicity were a very heterogenous group, which may explain why no clear pattern in PA determinants was observed in our results. However, the combination of high current physical literacy scores and a small number of PA types may indicate high levels of confidence participating in a few, ethnicity-specific PA types [45]. Interestingly, our results displaying that "Other" ethnicities conduct PA in the highest number of settings also contrasts previous work, given that minority ethnic groups are usually depicted as having little access to different PA settings [46].

The low PA determinant scores observed in disabled versus non-disabled adolescents can be explained through the social model of disability [47,48]. The model explains that people are not disabled because of their conditions, but instead due to the limitations of society to account for their disabilities. For example, lack of physical access to recreation facilities and specialised equipment availability can limit any PA engagement from disabled adolescents [49]. This can lower their confidence and motivation for participating in future PA, explaining the low physical literacy scores we observed. Furthermore, disabled people may experience a perceived or actual lack of knowledge or acceptance from PA facilitators such as coaches or recreation facility employees, culminating in little overall social support [49,50].

Higher deprivation levels are a marker of poor socioeconomic status for adolescents, which has previously been shown to negatively influence PA participation [51,52]. We observed low physical literacy scores for adolescents in higher deprivation neighbourhoods, possibly due to issues with accessing facilities that would develop fundamental PA skills and autonomy [53]. For example, higher deprivation neighbourhoods are often located further away from many recreational facilities, resulting in high transport costs and limiting the time adolescents can spend in a diverse range of PA environments [52]. Additionally, education facilities may be limited and have few PA resources available, reducing the support for adolescents in higher deprivation neighbourhoods to engage in physical activity and develop physical literacy [54]. However, the relative consistency in the results for adolescents living in high and medium deprivation areas is inconsistent with previous research and warrants further investigation (see Section 4.2) [55].

### 4.1. Implications

Achieving equity in PA participation among sociodemographic groups and finding all-encompassing methods to achieve this is paramount for enhancing wellbeing on a national scale. However, our results reflect that a ’one size fits all’ approach to addressing the determinants of future PA participation is unlikely to be successful given the differing strengths and needs across sociodemographic groups. Instead, efforts to promote PA should be wide-ranging and tailored to address the key determinants of participation identified for different groups of people. This is inherently complex, but one approach that does enable flexibility according to the needs of different population groups is the support of locally led initiatives. These often take a strengths-based approach and have been shown to enable the development of highly effective and sustainable interventions that provide autonomy and resources for people to address their own PA needs [56,57].

It could be assumed that sociodemographic groups with better PA determinant scores, such as the Māori and Pacific adolescents within our study, would perform better in other wellbeing measures. However, previous work displays the failings of national systems that have resulted in poor wellbeing for these population groups [58,59]. These results reiterate that high PA participation alone may not result in positive wellbeing and that PA recommendations must be made while considering the quality of PA experiences as well as broader personal preferences and life situations. Furthermore, this juxtaposition highlights the importance of considering strength-based and culturally appropriate approaches for increasing PA participation and variety across many populations [60].

Finally, this study highlighted the importance of assessing a wide range of PA determinants that represent different levels within the socio-ecological model. Intervention recommendations are strengthened when we retroactively consider how behaviour is influenced by interactions both with and around an adolescent. However, the data we had available still limits our ability to encapsulate higher-level interactions such as the sociocultural, physical, and policy environments surrounding the adolescent [61]. Therefore, work is needed on how we assess and collate data across the full array of socio-ecological determinants of adolescent PA behaviour.

### 4.2. Limitations

Existing evidence has described the impact that PA participation has on many domains of wellbeing [62,63,64] in addition to the possibility of a reciprocal relationship between PA and wellbeing [65]. However, our study is limited to only examining current determinants of future PA participation and does not include any measures of broader wellbeing. Future work in this field could look at including one or many proxies of wellbeing alongside determinants of future PA to further explore the relationship between these aspects.

Another limitation is that the cross-sectional nature of our data means that we can only predict, rather than definitively state, how current determinants affect long-term future PA participation in adolescents. However, previous research has clearly identified several of the key determinants of future PA participation included in our study [12,13,16]. Despite this, more longitudinal studies that track adolescents over time into adulthood are needed to further confirm the validity of our cross-sectional findings and more clearly identify what determinants of future PA participation experts should focus on.

Individuals from certain sociodemographic groups were underrepresented in our sample. For example, in our study, there are a lower proportion of adolescents with high deprivation, or who were gender diverse, compared to low/medium deprivation and male/female, which may explain why such wide CI estimates were observed and could have introduced some bias into the results we observed [29]. To attenuate this issue, we utilised a large sample size and used weighting to help bring our sample demographics closer to the national level. However, some sociodemographic groups have rarely been collected, such as gender diverse in the last national census, resulting in unreliable population numbers on which to base our weights [29]. Future work should look at data collection options that stem from collaboration across multiple governmental sectors, allowing access to a larger variety of datasets and promoting representation from these groups.

## 5. Conclusions

The current state of PA determinants in adolescents appears to be distinctly different between sociodemographic groups. A particular focus on addressing the determinants of PA participation is warranted within adolescents who are older, Asian, gender diverse, physically disabled, and from medium to high deprivation neighbourhoods. Finding ways to enhance multiple PA determinants during adolescence may greatly benefit future PA participation and subsequent wellbeing. Therefore, further investigation into exploring PA intervention options that affect different adolescent population groups is warranted, as is establishing longitudinal surveillance measures of PA determinants and participation across the lifespan.

## Data Availability

Publicly available datasets were analysed in this study. These data can be provided on request by contacting research@sportnz.org.nz.

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
