# Peer review of "Determinants of Future Physical Activity Participation in New Zealand Adolescents across Sociodemographic Groups: A Descriptive Study"

_ijerph, 2023, doi:10.3390/ijerph20116001_

Round 1

Reviewer 1 Report

The manuscript is addressed on an interesting topic but that is too familiar in the literature (determinants of future physical activity in adolescents across sociodemographic groups) and hardly adds anything new.

However, indicate that its writing and structure is clear, but I consider that its contributions are not enough to be published in a journal of such high quality as IJERPH.

Introduction Section is too short and poor.  For example, socio-ecological model is mentioned but is not explained. It is true that it is accompanied by a reference, but including more definitions and explanations in the manuscript would make it easier for the reader to read and understand. The same about the deprivation status variable. What was the type of response to give about the elements that make up the deprivation status variable? That is, for example, about income: are ranges proposed? What these ranges were? Or was the exact income requested? Or about Family Structure: what these different family typologies were? Although references are provided, offering more information makes it easier to read and understand the manuscript.

Line 263: Attention: “...determinants of disabled adolescent”. It has to say: ““...determinants of non-disabled adolescent”.

Author Response

Please see the attachment. Thank you for your work reviewing this paper.

Reviewer 2 Report

This manuscript aimed to explore the current state of physical activity participation in New Zealand adolescents expressed through various determinants of future physical activity across sociodemographic groups. The topic of the manuscript and study results are important for public health and physical activity promotion globally. The manuscript is well–written and easy to read.

There are my minimal comments:

The construction of the variable „Current PA availability issues“ is unclear. The authors state that it was constructed as the binary variable from the responses to two questions: 1) What activities have you participated in during the last 7 days; 2) What one activity would you do more of if you could? The second question was only asked of those who said they want to do more of PA that they did not currently participate in as having availability issues. It is unclear how the authors gathered this information (want to do more of PA that they did not currently participate in). Was there an additional question asking this? Please clarify it.

Please insert a sample size in the abstract.

Please mention in the abstract that the study is cross–sectional.

Author Response

(The authors gave the same response as above.)

Reviewer 3 Report

I do not think that in this article the comparative analyses would be enough for measuring only the descriptive statistical factors (or sociodemographic characteristics) such as gender and age, roots, or origins. I really miss emphasizing the results of qualitative factors and highlighting the role of social support more. In future research, I suggest measures of psychological factors as well, such as mental toughness, resilience, responsibility, and locus of control. According to the statistical analyses, maybe estimated marginal means would be more appropriate as a basis of comparison than just simply comparing cohort or gender group means.

Author Response

(The authors gave the same response as above.)

Round 2

Reviewer 1 Report

I believe that despite the changes introduced by the authors, the manuscript has not the sufficient quality to be published in IJERPH.  Its contributions are not relevant internationally.

The Introduction section is still too short and poor.
